# Influence of Nitrogen Fertilizer on the Antioxidative Potential of Basil Varieties (*Ocimum basilicum* L.)

**DOI:** 10.3390/molecules27175636

**Published:** 2022-09-01

**Authors:** Marzanna Hęś, Anna Golcz, Anna Gramza-Michałowska, Anna Jędrusek-Golińska, Krzysztof Dziedzic, Sylwia Mildner-Szkudlarz

**Affiliations:** 1Faculty of Food Science and Nutrition, Poznań University of Life Sciences, Wojska Polskiego 31, 60-624 Poznań, Poland; 2Faculty of Horticulture and Landscape Architecture, Poznań University of Life Sciences, Zgorzelecka 4, 60-198 Poznań, Poland

**Keywords:** basil cultivars, nitrogen fertilization, antioxidant activity

## Abstract

Total phenolic content (TPC) in extracts of basil depended on the cultivar and type of fertilization used in cultivation. TPC was determined spectrophotometrically with the Folin–Ciocalteu reagent. The antioxidant activity of extracts was analyzed by scavenging of DPPH and ABTS radicals, on the basis of metal chelating ability (MetChel) and ferric reducing antioxidant power (FRAP). The greatest TPC was determined in the purple cultivars—141.35 and 165.44 mg gallic acid/g d.m. for fertilized with ammonium nitrate (NH_4_NO_3_) and ammonium sulfate ((NH_4_)_2_SO_4_), respectively. Their extracts had the greatest antioxidant capacity in the majority of the methods used. The results varied depending on the modelling system used. The amount of polyphenols in individual basil cultivars differed significantly (*p* < 0.05) depending on the fertilization used in the culture. Regarding TPC, DPPH, ABTS, FRAP, and MetChel variables, we observed a significant effect for the applied cultivation. In the case of MetChel factor, lower results of all investigated basil species were observed for cultivation with ammonium sulfate. PCA demonstrated in the present study shows that Sweet and Cinnamon Basil samples cultivated with ammonium nitrate create a separated group. We recommend cultivation with ammonium sulfate fertilizers for these varieties of basil. The high content of phenolic compounds demonstrated in Sweet and Cinnamon Basil cultivated with ammonium sulfate, and thus associated antioxidant activity, indicates that it can constitute a valuable source for bioactive compounds in a balanced diet.

## 1. Introduction

Medicinal plants constitute a valuable source of antioxidants. A high content of bioactive compounds, polyphenols, and essential oils means that they demonstrate anticancer, antimicrobial and anti-inflammatory effects [1]. Furthermore, the values of numerous herbaceous plants stem from their invaluable aromatic digestion-aiding properties, therefore they are frequently used as spices [2,3].

In recent years, it has been claimed that basil, apart from its basic nutritional function, exhibits other features as well, which have a positive impact on our health, protecting the body from the occurrence of numerous diseases. Antioxidants are linked with an active protective role, as they support the natural protective system of the body, ensuring correct oxidoreductive homeostasis [4].

Basil (*Ocimum*), classified in the sage family (*Lamiaceae*), is one of the best known medicinal plants [5]. The complexity of traits and considerable human interference (cultivation, hybridization, and selection) results in the fact that the taxonomic classification of genus *Ocimum* is difficult to define. The current divisions are mostly based on traits, which depend on environmental factors, including morphology and leaf color [6].

Basil herb is used as the raw material in herbal medicine. It contains i.e. phenolic compounds, essential oil, saponin compounds, terpenoids, alkaloids, steroids, and tannins [5,7]. The phenolic compounds are mostly dominated by: rosmarinic acid, cinnamic acid, ferulic acid, and caffeic acid [8]. It also includes eugenol, cirsilineol, isothymusin, isothymonin, orientin, and vicenin [9,10]. Thus far, published research indicates greater capacity for the synthesis of polyphenols by purple basils compared with green cultivars [7]. In addition, basil herb is a source of vitamin B1, C, niacin, chlorophyll pigments, and carotenoids and mineral components such as: Fe, Mn, Zn, Mg, Ca, P, K, and Na [7,11].

Active substances of basil herb mostly originate from essential oils. They constitute a multi-component mixture of compounds, mostly classified among mono- and sesquiterpenes [12]. The amount and chemical composition of the oil differs between individual cultivars [13]. Results of numerous studies indicate, however, that the typical predominant compounds of the essential oil from basil herb are: linalool, methylol chavicol, geraniol, 1,8-cineole, and eugenol [14].

The search for natural scavengers of free radicals as food additives and nutraceutical agents has intensified in the past decade because of increasing negative reactions to synthetic compounds by consumers. A high content of phytocompounds, in particular phenolic acids and flavonoids, causes basil extracts to demonstrate a considerable antioxidant activity. This antioxidant potential can be considerably increased by the essential oils present in basil [5,15]. In addition, circumstances exist indicating that the content of phenolic compounds of essential oil can be modified with soil fertilization [16]; thus, for the first time, we assess the influence of the type of nitrogen fertilizer on the content of total phenolic compounds and the antioxidative potential of extracts from various basil (*Ocimum basilicum* L.) cultivars. The results are compared with butylated hydroxytoluene (BHT) because it is a model antioxidant with well-documented antioxidant activity [17,18].

## 2. Results and Discussion

The average extraction efficiency was 13.4%. The highest efficiency was attained in the case of Purple Basil cultivated on a substrate fertilized with ammonium sulfate (16.5%) and the lowest for Cinnamon Basil growing on the same substrate (9.6%). A similar efficiency of alcoholic extraction of polyphenols was obtained in the study by Coelho et al. [19].

The conducted study demonstrated that the content of phenolic compounds in the ethanol extracts of basil depended on the cultivar and type of fertilization used in cultivation. The level of these compounds ranged between 20.34–165.44 mg GADE/g d.m. of extract (Table 1). Purple Basil was the best source of polyphenols. The average content of phenolic compounds was considerably higher than in the remaining samples (*p* < 0.05), and it remained at the level of 141.35–165.44 mg GADE/g d.m. The lowest content of phenolic compounds was found in Cinnamon Basil A (20.34 mg GADE/g.d.m). Bajomo et al. [8] studied the total phenolic compounds in 22 cultivars of basil in a greenhouse and reported values from 3.07 to 4.99 mg GAE/g d.m. Total phenolic contents were statistically different among basil morphotypes within this study. Differences in the phenolic compounds’ content depend on different basil varieties cultivation and clime condition. Thus far, published research also indicates a greater capacity for the synthesis of polyphenols by Purple Basils compared with green cultivars [7]. As stated by Snežana [5] and Burdina and Priss [7], basil cultivars with purple coloration are characterized by a greater pool of polyphenols largely due to the presence of considerable amounts of anthocyanins. The level of anthocyanins is determined by the genotype of the given plant, but it may be subject to environmental modifications. Saadatian et al. [20] extracted 100 mg of GADE/g d.m. from leaves of red basil cultivars. The literature data indicates the importance of nitrogen dose. In the majority of cases, the higher nitrogen supply is linked to lower synthesis of plant metabolites, including polyphenols [21,22]. The content of phenolic compounds in green leaf cultivars ranged between 20.34–129.59 mg GADE/g d.m. Among them, the greatest amount of polyphenols, significantly different from the remaining variants (*p* < 0.05), was determined in Sweet Basil cultivated on the substrate containing ammonium sulfate. What is more, the study demonstrated that the same basil cultivated on a different substrate, where ammonium nitrate (NH_4_NO_3_) was the source of nitrogen, contained a significantly lower amount (*p* < 0.05) of polyphenols compounds among the analyzed cultivars. A similar relationship was found for the Cinnamon Basil. The amount of polyphenols in individual basil cultivars differed significantly (*p* < 0.05) depending on the fertilization used in the culture. For Greek and Lemon Basil, substrate fertilized with NH_4_NO_3_ turned out to be more favorable, while for Purple Basil, similarly to the aforementioned Cinnamon and Sweet Basil, it was the substrate enriched with (NH_4_)_2_SO_4_. Regarding the two different forms of nitrogen ((NH_4_)_2_SO_4_ and NH_4_NO_3_), higher phenolic compound content in cultivars with ammonium sulphate fertilizer was observed in the case of Sweet Basil. In Prinsi et al.’s [23] research, plants of the cultivars ‘Italiano Classico’ (green) and ‘Red Rubin’ (purple) were grown in hydroponics and subjected to different nutritional treatments, consisting of N starvation and nitrate (NO_3_^−^) or ammonium (NH_4_^+^) nutrition. This study reveals that N starvation, as well as the availability of the two inorganic N forms, differently affect the phenolic composition in the two cultivars. Compared with plants grown in NO_3_^−^ availability, in NH_4_^+^ availability, green basil showed a higher content of (poly)phenolic acids, while in purple basil, an increase in the contents of anthocyanins was detected. Overall, the study suggests that the management of NH_4_^+^ supply could contribute to the enhancement of the crop quality in hydroponics and provide new knowledge about the relationship between N nutrition and phenolic metabolism in basil.

In the research of radical scavenging activity (RSA), the capacity for DPPH^•^ scavenging by extracts of different basil cultivars was variable. The highest activity was demonstrated by Purple Basil extracts (192.2 mg Trolox/g d.m.) and Sweet Basil extracts (189.9 mg Trolox/g d.m.) fertilized with ammonium sulfate. However, this activity was approx. 61% lower than the synthetic antioxidant BHT (0.02%, *m*/*v*) (Table 1). Nguyen and Niemeyer [22] demonstrated that nitrogen fertilization also affected antiradical activity with basil treated at the highest applied nitrogen level, 5.0 mM, exhibiting significantly lower antioxidant activity than all other nitrogen treatments. The low antioxidant activities likely relate directly to the low total phenolic contents found for the same samples. Similarly, Cruz et al. [24] also observed that with a decrease in doses of nitrogen the antioxidant activity was higher. The limitation of fertilizer doses caused stress and resulted in flavonoids and phenolic substances synthesis.

The method using ABTS^•+^ demonstrated that significantly higher (*p* < 0.05) reducing potential was found for Sweet and Cinnamon Basil extracts fertilized with ammonium sulfate in comparison with ammonium nitrate. The scavenging capacity demonstrated by them was approx. 94% lower than the synthetic BHT antioxidant.

In the tested samples measured with the FRAP method, the most pronounced reduction in Fe^3+^ ions to Fe^2+^ was demonstrated by basil fertilized with ((NH_4_)_2_SO_4,_ which is contrary to other results obtained. The poorest reducing properties were demonstrated by Sweet and Cinnamon lavender cultivars also enriched with ammonium nitrate. Their capacity to reduce iron ions was 10-fold lower than BHT. The results are statistically significant (*p* < 0.05). The mean activity of the studied ethanol extracts of basil remained in the range between 3.1–29.5 mmol Fe^2+^/L. The reduction capacity estimated via FRAP method in the present study is considerably lower than stated by Salim et al. [25]. The mean activity of methanol extracts of basil studied by these authors was 66.3 mmol Fe^2+^/L. Various sources of nitrogen during plant growth can cause plant stress, which results in an increase in polyphenol content and antioxidant activity, which was demonstrated in earlier work [23]. This study reveals that N starvation, as well as the availability of the two inorganic N forms, differently affect the phenolic composition in the two cultivars.

All tested samples demonstrated the capacity to chelate metal ions in reaction with ferrozine. Among basil cultivars, the best efficiency in binding iron ions was found for the extract of Sweet Basil (fertilized with NH_4_NO_3_), which is contrary to the other acquired results. Type of fertilization did not affect the chelating activity for Greek Basil. The chelating capacity obtained by them was 90.9% and 97.8–98.1% poorer, respectively, as compared with BHT.

Regarding the mentioned analysis, the effect of different fertilizers type was observed, autonomous from varieties of basil. In order to check the effect of all variables (include varieties of basil), PCA analysis was performed (Figure 1). In the presented diagram, three groups of values are distinguished. The first group is Cinnamon and Sweet Basil cultivated with ammonium nitrate. The second group is remainder samples. The third group is BHT. Referring to all the variables conducted, only in the case of Cinnamon and Sweet Basil the significant effect of fertilization was observed. Nguyen and Niemeyer’s [22] results indicate that manipulation of nitrogen fertilization levels may be an effective method to increase the expression of polyphenolic compounds in basil. The content of total phenolic compounds in basil depends on the place and method of cultivation of the herb, the plant’s genetic conditions and method of herb drying, and its storage until analysis [20,26,27]. Taking into consideration individual results, we can firmly conclude that for Sweet and Cinnamon Basil ammonium sulfate fertilization is better (Table 2).

## 3. Materials and Methods

**Chemicals.** The following chemicals were used: 2,2-diphenyl-1-picrylhydrazyl (DPPH), 2,2′-azino-bis(3-ethylbenzothiazoline-6-sulfonic acid) (ABTS), ethylenediaminetetraacetic acid (EDTA), Folin-Ciocalteu reagent (FCR), 3-(2-pyridyl)-5,6-bis(4-phenylsulfonic acid)-1,2,4-triazine (Ferrozine), 2,4,6-tris(2-pyridyl)-s-triazine (TPTZ), gallic acid (GAD), (±)-6-hydroxy-2,5,7,8-tetramethylchromane-2-carboxylic acid (Trolox), and Butylated hydroxytoluene (BHT) and were obtained from Sigma-Aldrich (Darmstadt, Germany). Methanol, ethanol, hydrochloric acid, acetic acid, iron(III) chloride hexahydrate, sodium carbonate, and potassium persulfate were purchased from POCh (Gliwice, Poland). All chemicals and solvents used were of analytical grade.

**Plant material.** The basil leaves were obtained from the Marcelin Experimental Station of the Faculty of Horticulture and Landscape Architecture of the Poznań University of Life Sciences. The two-factor pot experiment involving cultivation of basil was carried out in a greenhouse in three replications. The experiment investigated the factors of the basil variety and the applied fertilizer (two type of nitrogen: NH_4_NO_3_ and (NH_4_)_2_SO_4_). In each replication, the experimental unit comprised 6 L boxes (2 plants per box; 2 plants × 5 varieties × 2 fertilizers × 3 replications, *n* = 60). The substrate in the culture consisted of peat limed to pH = 6.5. The moisture level in the substrate used in the cultivation of basil was maintained at 60% full water content (FWC). The substrate was limed on the basis of a neutralization curve, using the amount of 30 g of CaCO_3_/plant. Two types of nitrogen fertilizer were used in the experiment:(1)Ammonium nitrate NH_4_NO_3_ (designated in Table 1 variant A).(2)Ammonium sulfate (NH_4_)_2_SO_4_ (designated in Table 1 variant B).

Nitrogen dose was 120 mg/L. The nitrogen dose was determined experimentally. Based on the conducted research, it was found that this is the optimal dose of nitrogen to obtain a satisfactory yield of basil herb and allow the nitrate content to be kept below the permissible limit. The remaining macro and microcomponents constituted the study background. Growing phases were observed during the vegetation. Plants were harvested in full bloom. Leaves were separated from stems and freeze-dried. The characteristics of the chosen growth features of studied basil cultivars are presented in Table 2.

**Preparation of plant extracts.** Basil extracts were prepared by mixing 100 g of dried material with 300 mL of 80% ethanol (*v*/*v*) and triple-macerating overnight at room temperature. The environment of red-leaved varieties was additionally acidified with 0.1% HCl. The supernatants collected were filtered and ethanol was removed at 50 °C on a rotary evaporator. All samples were frozen at –20 °C for 24 h and then freeze-dried and stored at 4 °C in a dark place.

**Determination of total phenolic compounds.** The method using the FCR (Folin–Ciocalteu reagent) was used to determine the overall level of phenolic substances in the extract. A sample of the extract (0.2 mL containing 0.4–1.0 mg of extract) was mixed with 8.3 mL of distilled water and 0.5 mL of FCR. The resulting tested material was blended with 1 mL of saturated sodium carbonate solution. Subsequently, the mixture was incubated for 30 min at room temperature, resulting in a reading of the absorbance for 750 nm. The results are presented as mg gallic acid equivalent (GADE)/g d.m. extract.

**Scavenging of DPPH radicals.** DPPH^•^ (1 mM, 0.25 mL) was dissolved in pure methanol and added to 0.1 mL of extracts (1.0–4.0 mg/mL) or BHT (0.4 mg/mL) with 2 mL of methanol. The decrease in absorbance of the resulting solution was determined at 517 nm at 30 min. The results are presented as mg Trolox/g d.m. extract.

**Scavenging of ABTS radicals.** The radical cation was prepared by mixing 7 mM ABTS^•+^ stock solution with 2.45 mM potassium persulfate (1/0.5, *v*/*v*) and leaving the mixture for 12–16 h until the reaction was complete and the absorbance was stable. The ABTS^•+^ solution was diluted with ethanol to an absorbance of 0.700 ± 0.02 at 734 nm for measurements. The photometric assay was conducted on 3.0 mL of ABTS^•+^ solution and 0.03 mL of tested samples (0.5–2.0 mg/mL) or BHT (0.2 mg/mL) and mixed for 1 min; measurements were taken immediately at 734 nm after 6 min. The results are presented as mg Trolox/g d.m. extract.

**Metal chelating activity.** One milliliter of water solution containing 2–4 mg of extract or 1 mL ethanolic solution containing 0.2 mg of BHT was mixed with 3.7 mL of bidistilled water, and then the mixture was reacted with 0.1 mL of 2 mM FeCl_2_ and 0.2 mL of 5 mM ferrozine for 20 min. The absorbance was read at 562 nm. The results are presented as mg EDTA/g d.m. extract.

**Ferric reducing antioxidant power (FRAP).** FRAP reagent was prepared by mixing 300 mM acetate buffer, 10 mL TPTZ in 40 mM HCl, and 20 mM FeCl_3_ × 6H_2_O in the proportion of 10:1:1 at 37°C. A total of 3.0 mL of the FRAP reagent was mixed with 0.1 mL of the extracts at different concentrations (0.2–1.0 mg/mL) or BHT (0.2 mg/mL). The mixture was then incubated at 37 °C for 4 min in the dark. The absorbance was measured at 593 nm. The results are presented as mM Fe^2+^.

**Statistical analysis.** Statistical calculations using Tukey’s Honestly Significant Difference (HSD) test, box, and whisker plot were used for TPC, DPPH, ABTS, FRAP, and MetChel variables. Principal component analysis (PCA) was performed for observations of trends of data. All analysis was performed by Statistica software, ver. 10, StatSoft Inc. (Tulsa, OK, USA). Data was expressed as mean of 2 series and 3 independent measurements for each sample (*n* = 6).

## 4. Conclusions

Nitrogen fertilization and cultivar were determined to have a statistically significant effect on the total phenolic levels and antioxidant activities of basil. It should be emphasized that the assessment of antioxidative potential under in vitro conditions is an analysis which is difficult to interpret precisely and clearly. The action of antioxidants is based on different mechanisms; therefore, the methods verifying their activity are diverse. Additionally, taking into account all the analyzed variables, Sweet and Cinnamon cultivars grown in the presence of ammonium sulphate had the best antioxidant properties.

## Figures and Tables

**Figure 1 molecules-27-05636-f001:**
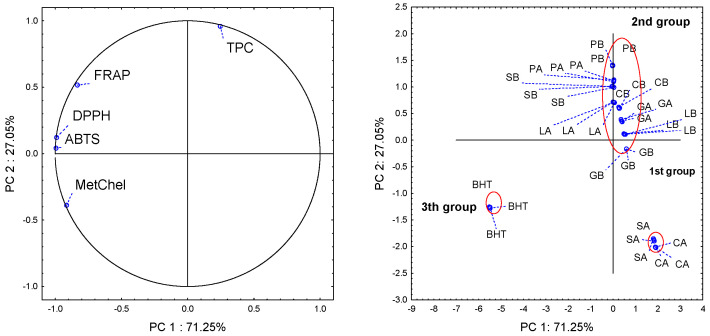
Principal component analysis for loading plot (TPC—total phenolic compounds, DPPH, ABTS, MetChel, and FRAP) and score plot (SA, SB, GA, GB, CA, CB, LA, LB, PA, PB, and BHT).

**Table 1 molecules-27-05636-t001:** Total phenolic content (TPC) and total capacity for extracts of basil cultivars determined using DPPH, ABTS, and FRAP tests and metal chelating activity.

Cultivar	TPC (mg GADE/g.d.m)		DPPH (mg Trolox/g d.m.)		ABTS (mg Trolox/g d.m.)		FRAP (mmol Fe^2+^/L)		Metal Chelating (mg EDTA/g d.m.)	
	Variant									
	A	B	A	B	A	B	A	B	A	B
green leaf basil										
Sweet	26.75 ^a^	129.59 ^e^	33.94 ^a^	189.85 ^g^	2.16 ^a^	11.29 ^de^	4.35 ^a^	27.67 ^g^	23.80 ^f^	6.14 ^b^
Greek	102.56 ^e^	80.45 ^b^	153.02 ^d^	123.10 ^b^	9.39 ^c^	9.35 ^c^	23.42 ^cd^	20.00 ^b^	4.83 ^a^	5.80 ^ab^
Cinnamon	20.34 ^a^	112.02 ^d^	26.61 ^a^	161.47 ^e^	2.26 ^a^	9.36 ^c^	3.09 ^a^	25.47 ^e^	19.49 ^e^	6.36 ^b^
Lemon	108.92 ^d^	90.33 ^c^	158.44 ^de^	131.66 ^c^	10.65 ^d^	8.44 ^b^	29.50 ^g^	22.89 ^c^	7.64 ^c^	6.47 ^a^
red leaf basil										
Purple	141.35 ^f^	165.44 ^g^	175.94 ^f^	192.17 ^g^	11.91 ^e^	13.40 ^f^	26.81 ^f^	24.52 ^de^	14.19 ^d^	19.49 ^e^
BHT			493.11 ^h^		36.45 ^g^		42.45 ^g^		260.72 ^g^	

Data are mean values, *n* = 6. Different letters indicate whether the difference is statistically significant given the method used (*p* < 0.05).

**Table 2 molecules-27-05636-t002:** Characteristics of chosen growth features for studied basil cultivars.

**Cultivar**	**Fertilization**	
Sweet Basil (*Ocimum basilicum)*	NH_4_NO_3_ (variant A)	(NH_4_)_2_SO_4_ (variant B)
Greek Basil (*Ocimum basilicum minimum*)		
Cinnamon Basil (*Ocimum basilicum cinnamon*)		
Lemon Basil (*Ocimum basilicum citriodora*)		
Purple Basil (*Ocimum basilicum purpurescens*)		
**Development phases**		
Sowing	8 May 2017	
Emerging	17 May 2017	
Harvest date	17 July 2017	

## Data Availability

The datasets generated during and/or analyzed during the current study are available from the corresponding author on reasonable request.

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
