# Peer review of "Influence of Nitrogen Fertilizer on the Antioxidative Potential of Basil Varieties (*Ocimum basilicum* L.)"

_molecules, 2022, doi:10.3390/molecules27175636_

Round 1

Reviewer 1 Report (New Reviewer)

GENERAL COMMENTS:

The authors present the results of the influence of nitrogen fertilizer on the antioxidant potential of different basil cultivars. In this work, the influence of the type of nitrogen fertilizer on the total phenolic compound content and antioxidant potential of extracts from different basil cultivars is studied. The results of the study show that nitrogen fertilization and cultivar have a statistically significant effect on total phenolic content and antioxidant activities of basil.

Overall, I consider the study relevant in the field.  In my opinion, however, authors should include some revisions (see my comments below). 

Lines 23-25: Must be amended based on later comments to indicate which samples of sweet and cinnamon basil form a separate group.

Lines 104 and 137: „ml“ shoud be „mL“. Please check throughout the whole text.

Lines 107 and 108: add space between number and . Please check throughout the whole text. Line 97: “type of nitrogen” rather than “types of nitrogen”.

Lines 97-98/146-148: Please describe more clearly the samples. It is not clear from these two sentences. “Each replica consisted of 20 plants, 10 for each types of nitrogen. Selected 15 random plants of boxes were used for recording the research.” “Data was expressed as mean ± standard deviation (SD) of 2 series and 3 independent measurements for each sample (n = 6). Where can these results be found?

Lines 133, 141, Table 2.: „mMFe2+/L“ should be „mmolFe2+/L“or „mMFe2+“, Please check throughout the whole text.

Table 2. (mgGAE/g.d.m) should be (mgGADE/g.d.m). Fig.1: Poor resolution, needs improving.

Line 227: The preceding discussion needs improvement in some respects. Confidence ellipses and loadings should also be included in Figure 2 (probably as a biplot so that scores and loadings are in the same plot ).

What about other class statistics arising from the confusion matrix such as selectivity, sensitivity, error, precision?

References 

Check for formatting consistency and errors.

Author Response

Dear Reviewer, I would like to thank you for your valuable comments, which could improve our manuscript.

The answer was placed below:

Abstract: Regarding to your comments we pointed which sample of basil create separate group: PCA demonstrated in the present study shows that Sweet and Cinnamon Basil samples cultivated with ammonium nitrate create separated group.

“ml” was corrected within all text for “mL”

Line 107-108 spaces were added.

Line 97: “types” was changed for “type”

Line 97-98 and 146-148: Thank you for your observation. We corrected the information in Material and method section and “Statistical analysis” section”: .

The basil leaves were obtained from the Marcelin Experimental Station of the Faculty of Horticulture and Landscape Architecture of the Poznań University of Life Sciences. The two-factor pot experiment involving cultivation of basil was carried out in a greenhouse in three replications. The experiment investigated the factors of the basil variety and the applied fertilizer (two type of nitrogen: NH4NO3 and (NH4)2SO4). In each replication the experimental unit comprised 6 L boxes (2 plants per box; 2 plants x 5 varieties x 2 fertilizers x 3 replications, n=60). From May 8 to July 17, 2017, five cultivars of basil have grown in an unheated greenhouse, in 6 L boxes (2 plants per box). The substrate in the culture consisted of peat limed to pH = 6.5. The moisture level in the substrate used in the cultivation of basil was maintained at 60% full water content (FWC).The substrate was limed on the basis of neutralization curve, using the amount of 30 g of CaCO3/plant. Two type of nitrogen fertilizer were used in the experiment:

  • ammonium nitrate NH4NO3 (designated in Table 1 variant A)
  • ammonium sulfate (NH4)2SO4 (designated in Table 1 variant B)

Nitrogen dose was 120 mg/L. The nitrogen dose was determined experimentally. Based on the conducted research, it was found that this is the optimal dose of nitrogen to obtain a satisfactory yield of basil herb and allows the nitrate content to be kept below the permissible limit. The remaining macro and microcomponents constituted the study background. On June 8, basil seedlings at the stage of 4-5 leaves and 12-15 cm in height were planted into boxes. Each replica consisted of 20 plants, 10 for each types of nitrogen. Selected 15 random plants of boxes were used for recording the research. Growing phases were observed during the vegetation. Plants were harvested in full bloom. Leaves were separated from stems and freeze-dried. Characteristics of chosen growth’s features of studied basil cultivars are presented in Table 1.

Statistical analysis line 146-148 concern prepared extracts of all basil samples. The obtained results were shown in table 2.

Table 2: line 133-141 mM” was changed for “mmol” within all text.

Table 2, “mgGAE” was changed for “mgGADE”

Fig.1 was deleted.

Line 227: discussion was modified. Confidence ellipses and loadings is not possible to place in one figure, because the Statistic software have some limitations. Therefore we did not change the figure 2. Selectivity, sensitivity and precision is important in the case of HPLC analysis, GC and others. Regarding to DPPH, FRAP, and other methods presented in this manuscript we did not estimate this parameters. For methods, only the mean and standard deviation are reported.

References: it was checked formatting consistency and errors.

Reviewer 2 Report (Previous Reviewer 1)

The paper submitted to Molecules fits partially within the scope of the Journal and it is much more suitable for an agronomical or horticultural journal such as Agronomy, Agriculture or Horticulturae.

I have several major concerns related to the novelty the robustness of the results and conclusions. The authors assess the effects of 5 cvs of basil subjected to two fertilization regimes ammonium nitrate versus ammonium sulfate.

1) It is not clear how the application of the two fertilizers were normalized. The number of replicates and number of plants per treatment is not clear.

2) What will happen if the experiments were carried out under different environmental conditions like different growing seasons all these qualitative parameters will be changed.

3) the data set of this experiment 2 figures e 2 tables which can be merged in just one table and 1 figure could not justify publication.

4) The discussion is very weak it is not sufficient to indicate whether the results are in line or not with the scientific literature but why?

I could not recommend the publication in Molecules MDPI.

Author Response

Thank you very much for your comments and suggestions. See below text, we try to answer for all of your doubt:

Regarding to climate change, new possibilities of breeding and cultivating medicinal plants in order to maintain or increase their content of bioactive substances should be searched for. Therefore, the proposed research is interdisciplinary and partly fits the subject of the Molecules journal.

  • Application of the fertilization was normalized regarding to previously experiments, where the authors suggested from 100 till 600 ppm of fertilizers (doi: 10.35248/2167-0412.20.9.345). The number of replicates and number of plants per treatment were corrected:

The basil leaves were obtained from the Marcelin Experimental Station of the Faculty of Horticulture and Landscape Architecture of the Poznań University of Life Sciences. The two-factor pot experiment involving cultivation of basil was carried out in a greenhouse in three replications. The experiment investigated the factors of the basil variety and the applied fertilizer (two type of nitrogen: NH4NO3 and (NH4)2SO4). In each replication the experimental unit comprised 6 L boxes (2 plants per box; 2 plants x 5 varieties x 2 fertilizers x 3 replications, n=60). From May 8 to July 17, 2017, five cultivars of basil have grown in an unheated greenhouse, in 6 L boxes (2 plants per box). The substrate in the culture consisted of peat limed to pH = 6.5. The moisture level in the substrate used in the cultivation of basil was maintained at 60% full water content (FWC).The substrate was limed on the basis of neutralization curve, using the amount of 30 g of CaCO3/plant. Two type of nitrogen fertilizer were used in the experiment:

  • ammonium nitrate NH4NO3 (designated in Table 1 variant A)
  • ammonium sulfate (NH4)2SO4 (designated in Table 1 variant B)

Nitrogen dose was 120 mg/L. The nitrogen dose was determined experimentally. Based on the conducted research, it was found that this is the optimal dose of nitrogen to obtain a satisfactory yield of basil herb and allows the nitrate content to be kept below the permissible limit. The remaining macro and microcomponents constituted the study background. On June 8, basil seedlings at the stage of 4-5 leaves and 12-15 cm in height were planted into boxes. Each replica consisted of 20 plants, 10 for each types of nitrogen. Selected 15 random plants of boxes were used for recording the research. Growing phases were observed during the vegetation. Plants were harvested in full bloom. Leaves were separated from stems and freeze-dried. Characteristics of chosen growth’s features of studied basil cultivars are presented in Table 1.

  • Taking into consideration previously papers we agree that different growing seasons can changed the investigated parameters, therefore important is to applying outline ready described. Basil is specific in this respect, as has been observed and described by many authors (https://doi.org/10.1155/2020/3808909; doi:10.3390/su11236590; Journal of Medicinal Plants and By-products (2019) 2: 143-151).
  • Figure 1 was deleted. Fig. 2 showed all of obtained results (240 dates) in one dimension by Principal Component Analysis used.
  • Discussion was corrected and improved. Some mistakes of used fertilizers were studied again and all confused information were deleted.

Reviewer 3 Report (New Reviewer)

The article is interesting, but it does contain some inaccuracies. They concern:

title: should be a bit shorter, in its current form it is quite illegible.

This is also related to the suggestion for keywords, which are too many and do not fully reflect the problem of the topic. In addition, these should be phrases, the most important keys that the reader can use to find the article in search engines. The authors are requested to correct these inaccuracies.

Introduction: It's pretty good. The authors drew attention to important issues that introduce the reader to the sense of the conducted experiment in relation to important research concerning the scope of the topic.

However, the authors did not indicate the novelty of their experiment in relation to the results known from the literature. How does the currently presented research differ from the previous ones that have already been published? Therefore, please outline the background around the analyzed raw material, especially since the authors later provide detailed information on the characteristics and occurrence as well as their possible applications.

Chapter: Conclusions

The authors are asked to formulate 2-3 important conclusions that result from the conducted experiment. Please also indicate the practical potential of the conducted research.

Chapter: References

Although the authors reached for many reports thematically related to the described results of the experiment, it should be clearly stated that these are quite old reports. In the available international literature there are many more recent reports related to the subject of the manuscript. Therefore, the authors need to refresh the literature a bit, so that it is not only historical, but also indicates the latest reports. Please also refer to the research conducted in the climatic zone in which the authors conducted their research - Eastern Europe, Polish National Centers. This is important as it indicates an important aspect related to the cultivation of basil.

Author Response

Thank you for valuable comments. All answers are placed below:

The title: The title of the manuscript was corrected regarding to your comments:

Influence of nitrogen fertilizer on the antioxidative potential of basil varieties (Ocimum basilicum L.)

Keywords:  the keywords were change regarding to your suggestion: basil cultivars, nitrogen fertilization, antioxidant activity.

Regarding “Introduction” we placed new sentence about investigated raw material: The search for natural scavengers of free radicals as food additives and nutraceutical agents has intensified in the past decade because of increasing negative reactions to synthetic compounds by consumers.

We also pointed that our experiment is novel:

In addition, circumstances exist indicating that the content of phenolic compounds of essential oil can be modified with soil fertilization [16], thus for the first time we the aim of the study was to assess the influence of the type of nitrogen fertilizer on the content of total phenolic compounds and the antioxidative potential of extracts from various basil (Ocimum basilicum L.) cultivars. The results were compared with butylated hydroxytoluene (BHT), because it is a model antioxidant with well-documented antioxidant activity [17,18].

Conclusion: The section was improved regarding to your comments:

The nitrogen fertilization and cultivar was determined to have a statistically significant effect on total phenolic levels and antioxidant activities of basil. It should be emphasized that the assessment of antioxidative potential under in vitro conditions is an analysis, which is difficult for a precise and clear interpretation. The action of antioxidants is based on different mechanisms, therefore the methods verifying their activity are diverse. The high content of phenolic compounds demonstrated in the present study and thus associated antioxidant activity indicates that basil can constitute a valuable source for bioactive compounds in a balanced diet. Additionally, taking into account all the analyzed variables, Sweet and Cinamon cultivars grown in the presence of ammonium sulphate had the best antioxidant properties.for cinnamon and sweet basil cultivation, regarding to activities of bioactive substances, we recommend the application of ammonium sulfate in comparison to ammonium nitrate.

Literature: The literature was changed for more actual positions: 6, 8, 9, 10, 15, 17, 19, 23, and 24 were replaced. New European literature was also added. Unfortunately we did not find right position described basil in Poland and around.

Round 2

Reviewer 2 Report (Previous Reviewer 1)

the revised version has been improved the manuscript can be accepted for publication in Molecules MDPI.

This manuscript is a resubmission of an earlier submission. The following is a list of the peer review reports and author responses from that submission.

Round 1

Reviewer 1 Report

The topic of the current manuscript fit partially with the scope of the Journal and from my point of view it fits better in a plant-oriented Journals such as Plants, Agriculture, Agronomy or Horticulturae MDPI. I have several concerns related to the originality, the robustness of the methods used and the conclusions. The agronomical parts in the materials and methods is completely missing, the authors did not report clearly the experimental design, the number o experimental units or replicates as well as the number of plants per replicate. From my point of view, the treatments should be analyzed in a factorial design with two treatments the 5 cultivars of basil and from the other side the two fertilization regimes. Even the fertilization regimes are not very well explained which make the conclusions very poor. In addition, the authors did not find any significant differences between the two forms of nitrogen so it seems a simple variety trial and this is not new since several experiments and published articles on the phenolic and aromatic profile of basil is reported in the scientific literature. The discussion section is very poor and descriptive since the authors failed to explain the differences between the 5 cultivars and this could be expected since the data set is quite limited. Almost all the analysis was conducted sepcrtophometrically and not using High quality instruments such as Orbitrap or HPLC. Based on the above considerations I could not recommend the publication in Antioxidants MDPI.

Reviewer 2 Report

The present manuscript aimed to investigate the influence of two nitrogen fertilizers (NH4NO3 and (NH4)2SO4) on the total phenolic content (TPC) and antioxidant potential of various basil cultivars. However, no clear differences were observed after treatments with both types of nitrogen fertilizers on the TPC and antioxidant activity of basil hidroethanolic extracts.

In my opinion, it would be important to analyze individual compounds in the extracts by HPLC-MS, for example, in order to infer about the impact of fertilizers in phenolic composition and provide sufficient advance about the impact of fertilizers in antioxidants content. In this case, authors based their conclusions only in non-specific colorimetric assays. Moreover, the results should be compared with basil cultivars without treatment in order to evaluate the impact of nitrogen fertilizers. In my opinion, control treatments without fertilizers would be of extremely importance. Also, authors only compare ethanolic extracts from basil cultivars, when it was previously described and highlighted in introduction that essential oils also contain important active compounds that can be affected by these treatments.

For these reasons, the manuscript should be improved in order to be published in antioxidant.

Remarks in detail:

Page 2, line 45-49: Authors should review bibliography and include more information about the content of individual phenolic contents and TPC in basil.

Page 2, line 62: Why the comparison with BHT?

Page 2, line 64: Information about water purity used for plants treatments and analytical assays should be indicated.

Page 2, line 81: How was defined the nitrogen dose?

Page 3, table 1: I suggest authors to indicate and distinguish green leaf cultivars from purple cultivars.

Page 3, line 87: Please substitute cm3 for mL and indicate 80% ethanol (v/v).

Page 3, line 100: The plant extract used for TPC determination was ressuspended from the freeze-dried residues? In which solvent? The preparation the extract should be clearly described because it is important for quantification purposes.

Page 6, table 2. Please use g d.m instead of g.d.m. Correct along all the manuscript.

Page 11, line 338: Why not to test other solvents? If polar compounds are dominant it would be preferable to use higher water proportions.

Page 12, conclusion: The conclusions regarding the impact of extracts on in vivo mechanism and diseases prevention must be supported by proper studies. Also, the analysis of individual constituents on the ethanolic extracts could allow to obtain more information about the impact of fertilizers treatments on individual antioxidants and bioactive phenolic compounds. The conclusion section must be reviewed.

Round 2

Reviewer 1 Report

I am sorry again to reject the paper. My main concerns remain the same that I indicate during the first round of revisions. Anyhow the final decision will be the one of the handling editor.

Bests

Reviewer 2 Report

In my opinion, the manuscript was significantly improved but still lacks the analysis of individual compounds in the extracts by HPLC-MS. Also, the absence of control plants without nitrogen fertilization was not clearly enlightened.As indicated by the authors and previously described by Nguyen and Niemeyer’s, manipulation of nitrogen fertilization levels may be an effective method to increase the expression of polyphenolic compounds in basil, with a significant impact on specific phenolic classes. For this reason, it is extremely important to understand the behavior of individual antioxidants induced by treatments.

Based on the above considerations, the submitted results do not provide new advances about the impact of fertilizers on antioxidants in basil and I could not recommend the manuscript for publication in Antioxidants.